# Single Mutations in Cytochrome P450 Oxidoreductase Can Alter the Specificity of Human Cytochrome P450 1A2-Mediated Caffeine Metabolism

**DOI:** 10.3390/biom13071083

**Published:** 2023-07-06

**Authors:** Francisco Esteves, Cristina M. M. Almeida, Sofia Silva, Inês Saldanha, Philippe Urban, José Rueff, Denis Pompon, Gilles Truan, Michel Kranendonk

**Affiliations:** 1ToxOmics, NOVA Medical School, Faculdade de Ciências Médicas, NMS|FCM, Universidade NOVA de Lisboa, Campo Mártires da Pátria 130, 1169-056 Lisbon, Portugal; ines.saldanha@nms.unl.pt (I.S.); jose.rueff@nms.unl.pt (J.R.); 2iMed.UL (Institute for Medicines and Pharmaceutical Sciences, Portugal), Faculty of Pharmacy, University of Lisboa, Av. Prof. Gama Pinto, 2, 1649-003 Lisbon, Portugal; calmeida@ff.ulisboa.pt; 3Laboratory of Bromatology and Water Quality, Faculty of Pharmacy, University of Lisbon, Av. Prof. Gama Pinto, 2, 1649-003 Lisbon, Portugal; sofia.silva3@campus.ul.pt; 4TBI, Université de Toulouse, CNRS, INRAE, INSA, 135 Avenue de Rangueil, 31077 Toulouse, CEDEX 04, France; urban@insa-toulouse.fr (P.U.); dpompon@insa-toulouse.fr (D.P.); gilles.truan@insa-toulouse.fr (G.T.)

**Keywords:** cytochrome P450, reductase, protein dynamics, enzyme mechanism, enzyme mutation, caffeine metabolism

## Abstract

A unique cytochrome P450 (CYP) oxidoreductase (CPR) sustains activities of human microsomal CYPs. Its function requires toggling between a closed conformation enabling electron transfers from NADPH to FAD and then FMN cofactors and open conformations forming complexes and transferring electrons to CYPs. We previously demonstrated that distinct features of the hinge region linking the FAD and FMN domain (FD) modulate conformer poses and their interactions with CYPs. Specific FD residues contribute in a CYP isoform-dependent manner to the recognition and electron transfer mechanisms that are additionally modulated by the structure of CYP-bound substrate. To obtain insights into the underlying mechanisms, we analyzed how hinge region and FD mutations influence CYP1A2-mediated caffeine metabolism. Activities, metabolite profiles, regiospecificity and coupling efficiencies were evaluated in regard to the structural features and molecular dynamics of complexes bearing alternate substrate poses at the CYP active site. Studies reveal that FD variants not only modulate CYP activities but surprisingly the regiospecificity of reactions. Computational approaches evidenced that the considered mutations are generally in close contact with residues at the FD–CYP interface, exhibiting induced fits during complexation and modified dynamics depending on caffeine presence and orientation. It was concluded that dynamic coupling between FD mutations, the complex interface and CYP active site exist consistently with the observed regiospecific alterations.

## 1. Introduction

Human cytochrome P450s (CYPs) form an enzyme superfamily containing 57 members with distinct cellular locations, being membrane-bound either in mitochondria or on the endoplasmic reticulum [1]. Fifty are microsomal, and are dependent on the electron supply from their obligatory redox partner cytochrome P450 oxidoreductase (CPR) [2]. This electron donation supports microsomal CYPs in metabolizing therapeutic drugs, and other xenobiotics, such as chemical carcinogens, environmental and food contaminants, but also endobiotic compounds, such as steroids, fatty acids, eicosanoids and vitamins [1,3]. Moreover, CPR is also the unique redox partner of structurally more divergent non-CYP enzymes, such as heme oxygenase [4] and squalene monooxygenase [5]. Due to this diversity of its redox partners, several of which play key roles in important metabolic pathways, CPR can be considered a central controller of many different physiological processes and cellular homeostasis, with a role to play in several pathophysiologies [6].

Human CPR, encoded by the *POR* gene, is a 79 kDa multidomain membrane-bound protein containing an N-terminal membrane anchor, a FMN-binding domain (FD) and a FAD-binding domain, which are separated by a connecting domain, and a C-terminal NADPH-binding domain [7]. The hinge region, a small flexible segment, connects the linker/FAD domains with the FD. For one CYP reaction cycle, two-electron transfer (ET) is necessary, occurring one at a time. This takes place through CPR’s transient interactions with microsomal CYPs, donating electrons (originating from NADPH), via intra-molecular transfer through CPR’s cofactors FAD to FMN, and subsequently via inter-molecular transfer from FMN to the heme moiety of CYP’s reactive center [2].

Conformational studies of CPR initially demonstrated closed structures of the protein, able to transfer electrons between the two flavin cofactors, but not to external redox partners [7,8,9]. Subsequently in 2009, three seminal studies described the existence of open conformations of CPR, compatible with ET to redox partners, which are demonstrative of CPR’s extensive protein dynamics in its ET function [10,11,12]. These structural studies demonstrated CPR’s transitions between closed and open conformers, enabled by the flexible connecting domain and in which the hinge region acts as a conformational axis for the swinging and partial rotational movement of the FD, relative to the rest of the protein [13,14,15]. In addition, this extensive protein dynamic was demonstrated to be highly dependent on ionic strength conditions [11,14,16], corroborating, at least in part, the longstanding knowledge of the effect of ionic strength (“salt effect”) on either cytochrome *c* (cyt *c*) or CYP reduction by CPR [17].

We previously reported on the essential roles of specific hinge segment residues in controlling the conformational equilibrium between the closed (for intra-molecular ET) and open (for inter-molecular ET) structures of CPR, namely residues G240, S243, I245 and R246 (numbering according to consensus sequence NP_000932.3) [18,19]. These residues seem to be critical although not exclusive, for electrostatic and flexibility properties of the hinge segment in CPR’s ET function [18]. Our data corroborate the earlier suggested “conformational sampling” of CPR by redox partners [14], in which the highly flexible hinge segment seems to be responsible for a large ensemble of open conformers which can be sampled for CYP isoform-specific interactions.

Microsomal CYPs do not present conserved key residues on their CPR binding surface at their proximal side for this interaction, compared to mitochondrial CYPs which contain conserved basic residues at their proximal surface for interaction with their electron donor adrenodoxin [20]. The diversity of surface residues displayed by microsomal CYPs implies a degenerated FD binding surface and affinity differences among microsomal CYP isoforms for CPR, a plausible key feature for the need for the sampling of the open conformers of CPR by structural diverse CYPs [19]. It seems therefore that an effective ET for CPR is enabled by (i) the existence of CPR’s extensive protein dynamics resulting in an aggregate of open conformers for sampling by its structural diverse redox partners and (ii) the affinity probing of these open conformers by CYP isoforms through the specific affinity characteristics for the FD of CPR.

The FD of CPR is highly conserved among mammals [21] and contains several conserved patches of acidic residues on its surface, which were formerly implicated in electrostatic interaction with redox partners (reviewed in [22]). Using random mutagenesis, we isolated FD mutants of CPR (e.g., P117H, G144C, N151D, G175D, H183Y and A229T) showing increased affinity and activity for one specific CYP isoform (CYP1A2, 2A6 or 3A4), while being either neutral or detrimental for the other two tested CYP forms [21]. Moreover, we demonstrated this CYP isoform-specific effect to be dependent on the structure of the substrate bound by the CYP [23], a phenomenon we previously observed with hinge mutants [19]. These specific FD mutations were found to be located in close vicinity or adjacent to (i) specific patches of acidic residues, formerly implicated in CPR–CYP interactions; (ii) natural occurring variants of the FD which were formerly demonstrated to cause CYP isoform effects (e.g., T142A, Q153R, V164M, D211N and P228L); (iii) tyrosine residues 143 and 181, both considered to be directly involved in the binding of the FMN moiety of CPR [21]. Molecular dynamics simulations demonstrated that several of these mutations induce very subtle structural modifications, altering the charge of the surface patches implicated in the CPR–CYP interaction and/or the fine positioning/surroundings of the FMN cofactor [21,23].

One may expect that the conformational plasticity of CYPs plays an additional role in the formation of the transient CPR–CYP complex for ET, as we previously noted [19]. The substrate-binding site of CYPs,, particularly those involved in drug metabolism, is highly malleable, and substrate binding induces subtle overall conformational changes [24], including alterations at the proximal site (reviewed in [25]). The manner of these conformational changes could be related to the proximal site and may influence the affinity of CYP for CPR and thus CPR sampling by CYPs. Rationalizations for the specificity of CYPs in generating specific metabolites are largely based on the three-dimensional architecture of the substrate-binding site and the compatible positioning of substrates relative to heme in the active site [26,27]. However, our results on hinge and FD mutants seem to support both CPR’s protein dynamics and the affinity sampling of the FD, as additional determinants in the specificity of microsomal CYP-mediated reactions.

The work herein describes our investigations pertaining to clarifying the roles of CPR’s open/closed dynamics and its FD in the metabolic outcome of CYP reactions. For this purpose, a set of the previously studied CPR mutants (both natural occurring as well as the created ones) of the FD and of the hinge region were used and each separately combined with human CYP1A2. To delineate their roles, we used caffeine as a probe substrate, an accepted phenotyping compound of human CYP1A2 [28]. CYP1A2 mainly produces three primary metabolites of caffeine in a specific profile of paraxanthine, theobromine and theophylline, namely 85, 10 and 5%, respectively [29]. Several of the studied CPR variants demonstrated altered profiles of these CYP1A2 metabolites when compared with the one obtained with the CPR wildtype, confirming the roles of CPR’s protein dynamics and its FD in the specificity of metabolite formation. Different experimental approaches, including computational ones, were used to unravel the molecular mechanisms underlying these observations.

## 2. Materials and Methods

Reagents L-Arginine, thiamine, chloramphenicol, ampicillin, kanamycin sulfate, isopropyl β-D-thiogalactoside (IPTG) (dioxane-free), δ-aminolevulinic acid, dichlorophenolindophenol (DCPIP), cytochrome *c* (horse heart), glucose 6-phosphate, glucose 6-phosphate dehydrogenase, nicotinamide adenine dinucleotide phosphate (NADP+ and NADPH), catalase (bovine liver), superoxide dismutase (SOD) (bovine erythrocytes), resorufin, methoxyresorufin (MthR), 2′,7′-dichlorodihydrofluorescein diacetate (DCFH2-DA), acetonitrile, DMSO, caffeine (98,5%) and paraxantin (98%) were obtained from Sigma-Aldrich (St. Louis, MO, USA). Theobromine (99%) and theophylline (99%) were purchased from Alfa Aesar (Kandel, Germany) and ACROS Organics (Thermo Fisher Scientific, Winsford, UK), respectively. Methanol (MeOH) was LC/MS purity-grade (JT Baker, Deventer, The Netherlands). Formic acid (liquid chromatography grade, ≥98%) and ammonium acetate (98%, p.a.) were obtained from Merck (Darmstadt, Germany). LB Broth, bacto tryptone and bacto peptone were purchased from BD Biosciences (San Jose, CA, USA). Bacto yeast extract was obtained from Formedium (Norwich, UK). A polyclonal antibody from rabbit serum raised against recombinant human CPR obtained from Genetex (Irvine, CA, USA) was used for the immune detection of the membrane-bound CPR. All other chemicals and solvents were of the highest grade commercially available.

In terms of CPR–CYP1A2 membrane fractions, the *E. coli* cell model BTC was used for the heterologous co-expression of human CYP1A2 together with full-length human CPR (either expressing WT, CPR mutants or no CPR), using a biplasmid co-expression system, as previously reported [18,19,21,30,31]. Membrane fractions of the different strains were prepared and characterized for total protein content (Bradford assay) as well as CYP (CO difference spectrophotometry) and CPR (immuno detection) contents, as described previously [19,30,32,33,34]. The intrinsic electron donation capacity of the FD mutants was assessed via DCPIP reduction, as previously described [18,21]. The assays were performed (N = 3) in a buffer containing 50 mM Tris, 150 mM KCl, 10 mM NaN3, and 0.04% Triton X-100, with the pH 7.5 at 37° C in the presence of 200 mM NADPH and 70 mM DCPIP. Initial rates were monitored at 600 nm using ΔεM = 21,000 M-1 cm^−1^. The functionality (stability) of CPRvar proteins during the 6h incubation period in the CYP1A2-mediated caffeine metabolism assay was assessed via the measurement of cyt *c* reduction activity at 0 and 6 h of incubation, using a 96-well plate format (SpectraMax^®^i3x, Molecular Devices, San Jose, CA, USA; SoftMax Pro 2.0), as previously reported [18]. Reactions were performed (N = 3), using 100 µM cyt *c*, NADPH regenerating system (NADPH 200 µM, glucose 6-phosphate 500 µM and glucose 6-phosphate dehydrogenase 40 U.L-1, all final concentrations), 1 nM CPR, catalase (0.2 U.µL-1) and SOD (0.05 U.µL-1), in a 100 mM Tris-HCl buffer containing 1 mM EDTA and 0.04% Triton X (pH 7.4). Reactions were followed at 550 nm, for 4 min (37 °C), and reduction velocity rates (Abs/min) were determined from the initial linear part of the absorption curve.

Regarding CYP1A2 enzyme activity, caffeine metabolism was evaluated by analyzing the formation of metabolites in a reaction mixture (200 µL) containing 600 µM caffeine (approximately 2 times the K_M_ concentrations of the three reactions mediated by CYP1A2 [35]) and 60 nM CYP1A2, in a 100 mM potassium phosphate buffer (pH 7.6) supplemented with 3 mM MgCl2 and a NADPH regenerating system (same as described above). After 6 h of incubation at 37 °C, reactions were quenched by adding 3 reaction volumes of iced cooled acetonitrile, mixed and incubated on ice for 10 min. Subsequently, the mixtures were centrifuged (4 °C, 21,000 g, 10 min), and supernatants were recovered and dried via vacuum centrifugation. The resulting dried material was reconstituted in water (200 µL) and contents of caffeine and metabolites were determined via ultra-performance liquid chromatography–tandem mass spectrometry (UPLC-MS/MS) [36,37].

The formation of ROS (i.e., hydrogen peroxide and superoxide) during the caffeine reaction was estimated separately using the 2′,7′-Dichlorofluorescein (DCFH_2_) oxidation assay [38]. DCFH_2_ was obtained from DCFH_2_-DA via deacetylation [39]. DCFH_2_ (10 µM, final concentration) was added to the reaction mixtures for CYP1A2-mediated caffeine metabolism, as described above. The formation of ROS was evaluated (N = 3) through a fluorescence measurement of formed DCF (excitation 495 nm; emission 525 nm), using a 96-well plate, for 6 h, at 37 °C. Values (FU/min) were corrected using identical reaction mixtures, without the NADPH regenerating system.

As for the cytochrome *c* reduction assay in the presence of caffeine, the cyt *c* reduction activity of CPR/CYP1A2 membrane fractions was determined using a 96-well plate format as described above [18], applying a caffeine gradient: 0–1000 µM (N = 3).

In terms of the UPLC-MS/MS analysis of CYP1A2-mediated caffeine metabolism, sample analysis was performed via ultra-performance liquid chromatography–tandem mass spectrometry (UPLC-MS/MS) using a Dionex Ultimate 3000 system coupled to a mass spectrometer, TSQ Endura triple quadrupole, from Thermo Scientific. A Kinetex EVO C18 column (2.1 cm × 50 mm × 2.6 μm) from Phenomenex (Torrance, CA, USA) was used to perform chromatographic separation. The tandem mass spectrometer operated in the positive ion ESI mode using the multiple reaction monitoring (MRM) mode. The chromatographic run involved the use of mobile phase A with a mixture of water, ammonium acetate (0.01 mM) and formic acid (0.5% *v/v*), and mobile phase B with 100% MeOH. The gradient program started with 95% of mobile phase A, followed by a linear decrease to 50% of mobile phase A until 3.0 min, 30% of mobile phase A until 5.5 min, and 10% of mobile phase A until 8.0 min (held 3.0 min). To re-equilibrate the system, an increase in mobile phase A to 95% was performed in 1.0 min (held 3.0 min). The injection volume was 20 µL, and the flow rate was 0.3 mL/min. Triple-quadrupole operating conditions were optimized to work in the multiple reaction monitoring mode (MRM). The optimized MS/MS conditions were based on the selection of the ionization mode, optimum collision energy (V), ion transfer tube, vaporizer temperatures (°C) and an accurate radio frequency (RF) lens for each compound. Nitrogen was used as a sheath, aux and sweep gas. Sheath gas flow was set to 40 Arb, and the aux gas flow was set to 10. Ion transfer tube and vaporizer temperatures were also optimized, with the selected temperature of 325 °C for the ion transfer tube and of 220 °C for the vaporizer (Appendix A). Stock solutions of standards were prepared, and MS/MS parameters were optimized for detection (see Appendix A on stock solutions and standards for UPLC-MS/MS, and Appendix A). Dried extracts were reconstituted with 200 µL of reagent water. For caffeine analysis, these reconstituted samples were diluted (20,000×) with reagent water. For metabolite analysis, dissolved residues were directly applied. Before caffeine chromatographic analysis, solutions were filtered with polytetrafluoroethylene (PTFE) syringe filters (13 mm × 0.45 μm, Millipore, Burlington, MA, USA).

In terms of the statistical analysis of membrane fraction data, variances in the data were analyzed through a two-way ANOVA or one-way ANOVA with Bonferroni’s multiple comparison test. The analysis was performed with a 95% confidence interval using GraphPad Prism 5.01 software (La Jolla, CA, USA). The significance level considered in all the statistical tests was 0.05.

### Computational Simulations

For the structural analysis of CRP mutants and CYP1A2, four methods were used, each applied according to the required outputs. Method A involved Alphafold2 (AF2A Advanced version) to generate a set of raw complex structures from the sequences of CYP1A2 and the FD of human reductase (excluding membrane anchor parts). Such an approach does not consider the potential induced fits involved occurring at the interface during complex formation. Five alternate complex structures were generated by AF2A. The one featuring the best interdomain affinity evaluated using the Prodigy web server was selected then relaxed using Rosetta Relax software. Method B extended method A by using the resulting A structure to seed RosettaDock-4.0 and calculate 1000 variants of this initial complex. The variant structures featuring the best I-SC scores combined with the best interdomain affinities were selected. In the case where the plot of I-SC scores as a function of the root mean square deviation (RMSD) of variants with the starting structure did not exhibit clear convergence, a second cycle of refinement was initiated involving Rosetta relaxation and Rosetta docking (up to three cycles). This iterative procedure allowed the progressive adjustment of the complex interface when induced fits were involved. Methods C and D are variants of method B involving seeding of the GalaxyRefineComplex software instead of RosettaDock during refinement cycles. The docking methods involved did not correctly manage the flavin and the heme cofactors, and cofactors were replaced on modeled complex structures at each refinement cycle via RMSD minimalization with individual reference crystallographic structures before the Rosetta relaxation steps. This was performed to consider potential contributions of the FMN and heme cofactors in modulating the adjustment of the side chains at the interface. Individual domains of finally modeled complexes exhibit RMSD values from 0.35 to 0.55 Å with PDB-1b1C (FD) and from 0.63 to 0.82 Å with PDB-2HI4 (CYP1A2). In comparison, RMSD values between crystallographic structures PDB-1c1c, 3qe2, 3qfc and 5fa6 ranged between 0.3 and 0.35 Å for FD. The affinities of complexes obtained with method C using Galaxy docking method 2 appeared higher by 2–4 kcal/mol than that for the complexes generated with RosettaDock due to the better-induced fits of residues in the linker part of the membrane anchor (residues 50–70 of the full-length sequence) close to the main FD fold. This difference was absent when using Galaxy docking method 1 and vanished when residues of this linker part were removed from affinity calculation. In modeled complexes, the distance between the flavin methyl groups and CYP iron ranged between 15 and 18 Å depending on the poses (13.5 to 17 Å for methyl groups to Cys thiol). Dihedral angles between FMN and heme planes ranged between 77 and 84° depending on the complex poses considered. Such geometries are consistent with the requirements for efficient electron transfers.

As for the identification of critical residues at the interface, thirteen different FD–CYP complex poses resulting from docking methods B and C were analyzed using a side chain contact distance equal to or lower than 3 Å as a criterion. A subset of involved residues was defined using the additional criterion that contact was detected in at least 50% of considered poses.

Regarding the induced fit RMSD calculation, the procedures used are depicted in the flow chart of Figure 1. Three independent representative poses of FD–CYP complex models were first submitted to aggressive relaxation and annealing using Rosetta software. Each relaxed structure was split into 3 derived structures: one for the isolated CYP, the second for the unchanged complex and the third for the isolated FD. Individual structures of the three sets of split models were each submitted to three independent new cycles of Rosetta relaxation and annealing giving a total of 27 (3 × 3 × 3) independently relaxed poses. Following the global alignment of all structures via RMSD minimization on their C-alpha, the RMSD values of side chains of individual residues in the complex and in separated domains were calculated. All possible combinations between members of the 27 structure sets were considered and similar combinations were averaged to improve the signal to random noise ratio. In all cases, residue side chain-specific RMSD values were normalized via division by the side chain atom count. Averaged side chain RMSD values resulting from the intradomain comparisons between relaxed poses were subtracted from the previously calculated RMSD values for the inter-model (complexed vs. free) comparison.

As for the structural analysis of CPR mutants, the study of three mutants (H183Y, T142A and V164M) was performed using the CPR FD only, obtained from the crystal structure of human CPRwt (PDB 5FA6) as the starting material. Mutations were constructed with the YASARA software (www.yasara.org). Fully hydrated molecular dynamics simulations were performed for 80 ns using a step size of 100 ps with the Yasara software suite using the AMBER14 force field at a constant pressure (water density at 0.997 g/mL) and temperature (298 K). Parameters were set as follows: cuboid cell extending 10 nm on each side of the protein, 0.9% NaCl and pH = 7.4. Electrostatic potentials were calculated on the average structures with the integrated module of the Pymol software (www.pymol.org) using default parameters (non-linear Poisson–Boltzmann equation; protein dielectric, 2.0; solvent dielectric, 78.0; temperature, 310 K; ion charge, +1 of radius 2.0 Å and −1 of radius 1.8 both at 0.15 M) and plotted on the solvent-excluded surfaces (Connolly surfaces).

With regard to docking to CYP1A2, structural analysis was performed using the crystal structure of human CYP1A2 (PDB 2HI4) as the starting material. All work was performed using Yasara Structure as the software suite [40]. Prior to all subsequent work, the structure was relaxed with the refined procedure of Yasara using the following parameters: 298 K, density of 0.997, pH of 7.4 and the force field of YASARA2 [41]. The subsequent relaxed structure was then used for docking. Caffeine docking was performed with the VINA procedure of YASARA [42] with an ensemble of CYP1A2-relaxed structures. The caffeine ligand was docked 15 times against each of the 20 receptors in the ensemble, yielding 300 complexes. After clustering the 300 runs, 138 distinct complex conformations were found. The three clusters corresponding to the putative orientations yielding the three major caffeine products with CYP1A2 were retained for molecular dynamics analysis. They are named the 1st, 2nd and 3rd position, compatible with the formation of theophylline, paraxanthine and theobromine, respectively.

In terms of the molecular dynamics of CYP1A2 with various caffeine poses, using the four different CYP1A2 starting structures (caffeine-void or bound in the first, second or third position), fully hydrated molecular dynamics simulations were performed for 25 ns using a step size of 100 ps with the Yasara software suite and the AMBER14 force field [43] at a constant pressure (water density at 0.997 g/mL) and temperature (298 K). Parameters were set as follows: cubic cell extending 5 nm on each side of the protein, 0.9% NaCl and a pH of 7.4. RMSD and root mean square fluctuation (RMSF) values were collected after each molecular dynamics run and analyzed with the R software suite. RMSF differences were calculated as follows. The average RMSF for each residue was calculated with either all atoms or backbone atoms only and averaged between the four simulations. These average values were then subtracted from the same values (backbone atoms or all atoms) from the average RMSF from caffeine-free simulations. Figures were prepared with the Pymol software and R ggplot function. All modeling data are available upon request.

## 3. Results

### 3.1. Isolation and Characterization of the Membrane Fractions

To study the effect of CPR’s protein dynamics and its FD in CYP1A2 caffeine metabolism, a total of 14 CPR variants (CPRvars) were each separately co-expressed with human CYP1A2, using the bi-plasmid *E. coli* cell model BTC [30,31,34]. Studies by our group and others demonstrated the role of specific residues of CPR, able to confer altered CYP activities in an isoform-dependent manner [6,32]. Considering this, the CPRvars evaluated in the present study were selected from our previous studies, targeting residues apparently part of a complex network controlling the open/close CPR dynamics and/or the CPR–CYP interaction modes. Namely, these were six created variants with mutated FD residues inducing isoform-specific augmented CYP activity (mutants P117H, G144C, N151D, G175D, H183Y and A229T) [21,23], five natural occurring variants with CYP isoform-specific effects (T142A, Q153R, V164M, D211N and P228L) [6,32] and three additional CPRvars interfering with structural properties of the hinge segment (S243, I245 and R246) [18,19]. Membrane fractions were isolated and characterized for their CYP1A2 and CPR contents (Table 1). The sub-molar presence of CPR vs. CYP in human liver microsomes [44,45] was properly recapitulated using the BTC system, as previously demonstrated in our former studies. When using this model, the co-expression of CPRvar with a specific CYP should be achieved with stoichiometries approximating the one obtained with CPRwt, demonstrating no significant differences, as properly demonstrated in our previous studies [19,21,30,33,46]. As CPRvar/1A2 membrane fractions demonstrated some discrepancy in the CPR:CYP ratios, three CPRwt/1A2 membrane fractions with distinct CPR:CYP ratios (1:2, 1:9 and 1:13) were used as references to correctly normalize the relative activities of the CPRvar/1A2 membrane fractions (Table 1). Overall, the FD mutants showed no apparent changes in the FMN content and/or redox potential, when compared to those of the CPRwt [21]. This was demonstrated by assessing the reduction rates of DCPIP in the created FD mutants, previously described in Esteves et al., 2021 [21], and in the five natural occurring variants (Appendix A). This assay measures the electron flow from NADPH to FAD and finally to FMN, which directly reduces DCPIP, a reaction independent of the protein–protein interaction involving the FMN domain interface. The only exception was found for variant T142A, for which the DCPIP reduction ratio was lower than that of the CPRwt, which may indicate the loss of FMN content and/or an alteration of redox potential. Our results are in accordance with those of a previous study, in which the total flavin content of CPRvar Q153R was comparable to that of the CPRwt, while variant T142A showed a significant loss of relative flavin content [47]. Thus, the data of CPRvar T142A were interpreted with caution. Protein stability differences among CPRvars during incubation may constitute an interfering factor. Stability was verified on the functional level by assessing cyt *c* reduction activity. No statistically significant alteration in cyt *c* reduction rates was observed for all variants studied, after 6 hrs of incubation vs. 0 hrs. of incubation, except for variant S243P (see Table 1). Therefore, any difference detected in CYP1A2 activity can be ascribed to deviations caused by the presence of CPR variants, except for CPRvar T142A and S243P, the data on which should be interpreted with caution (*vide supra*).

### 3.2. Caffeine Consumption

Membrane fractions containing CPR variants with CYP1A2 as well as respective controls (i.e., CPRwt/CYP1A2 and CPRnull/CYP1A2) were incubated with caffeine. Relative reaction velocities for each CPRvar/1A2 membrane fraction are presented in Figure 2. Overall, the kinetic parameter analysis evidenced that CYP1A2 activity was affected by the presence of CPR variants. CYP1A2 membrane fractions containing Q153R, D211N, H183Y, A229T or R246A showed augmented caffeine consumption, while P117H, G144G, N151D, G175D and particularly T142A and I245P demonstrated decreased activities, when compared to those of the CPRwt. Apparently, V164M, P228L and S243P have no effect on the caffeine consumption velocity rates mediated by CYP1A2.

### 3.3. Effect of CPR Variants on CYP1A2-Caffeine Metabolite Pattern

The relative caffeine metabolite profiles produced by the different CPRvar/1A2 membrane factions (i.e., vs. CPRwt) are shown in Figure 2 (see Table 1 for absolute metabolite contents). The reactions of the three CPRwt/1A2 couples, used as specific references, demonstrated relative contents of the three caffeine metabolites (86.0 ± 4.6% paraxanthine, 6.2 ± 1.0% theobromine and 7.8 ± 1.7% theophylline) as previously described [29]. Altered profiles of metabolites were observed, using the CPRvar/1A2 membrane fractions, when compared to those of the CPRwt. Several of these CPRvar/1A2 couples produced statistically different relative quantities of metabolites, namely those with CPR variants T142A, V164M, H183Y, A229T, S243P, I245P and R246A. These differences were mainly related to the relative proportions of theophylline and theobromine, which are reported as lower rate caffeine metabolites produced by human CYP1A2 [29,48].

### 3.4. Effect of CPR Variants on ROS Production in CYP1A2-Mediated Caffeine Metabolism

Initial structural analyses of several CPR variants which demonstrated significant differences in metabolite profiles (T142A, V164M and H183Y) showed subtle alterations on their surface electrostatic potential, near or in the FMN binding pocket (Appendix A). These may influence the redox potential of the bound FMN moiety and/or the level of uncoupling of the reaction. To probe the latter, ROS (i.e., hydrogen peroxide and/or superoxide) production rates during CYP1A2-mediated caffeine metabolism were estimated, applying the DCFH_2_ oxidation assay for all CPR variants and the CPRwt, and compared to the respective caffeine consumption rates (Appendix A).

Firstly, a relative low level of ROS formation was observed for all samples. Secondly, variants did not show increased ROS levels versus those of the CPRwt, except that variants T142A and A229T induced significant increased levels of ROS, compared to the other variants. Although the ROS production rate differed among variants, no direct correlation seemed apparent for the CPR variants with most deviated metabolite profiles (T142A, V164M, H183Y and S243P).

### 3.5. Cytochrome c Competition in the CPRvar/CYP1A2–Caffeine Reaction

As FD surface patches of CPR, involved in protein:protein interactions with both CYP and cyt *c*, apparently overlap [49,50], we previously applied cyt *c* competition studies to probe affinity differences among CPR variants and CYP [23]. To test the affinity changes of CPR variants for CYP1A2 metabolizing caffeine, we tested cyt *c* reduction in the presence of caffeine. Cytochrome *c* reduction velocities of variants T142A, V164M, H183Y and S243P (which demonstrated significant differences in metabolite profiles and/or in ROS production) were plotted in function of caffeine concentrations, see Figure 3 (see also Appendix A for reduction rates without caffeine). The observed velocity rates were not significantly inhibited by the increased caffeine concentrations, except for variant H183Y at the highest caffeine concentration. This might indicate an increase in the affinity of this variant for CYP1A2 interaction in the presence of caffeine, when compared to that of CPRwt and the other tested variants. However, incubations for metabolite detection (see above) were performed with 600 µM of caffeine and thus the observed effects of mutations seem to not have been due to an affinity difference. These results are in accordance with the corresponding ones on relative caffeine consumption (see Figure 2), in which H183Y showed the highest velocity rate, T142A demonstrated decreased, and V164M and S243P showed CPRwt-like velocities.

### 3.6. Interaction of CYP1A2 with CPR

In clarifying the described effects, we performed several modeling studies to understand the differential effects of CPR variants on the interaction with CYP1A2. A model was designed for the complex between the isolated FD of the reductase and CYP1A2 (see Material and Methods and Figure 1). Briefly, an initial complex geometry was generated using the Alphafold2_advanced version and respective sequences excluding membrane anchor parts. This initial complex, which lacked cofactors, was relaxed to remove possible structural clashes, and then used as a starting point to build the presented complex through several cycles of docking, cofactor replacement and relaxations to refine the initial complex and to allow the progressive adjustment of side chains at the domain interfaces. The final structure was generated and relaxed taking into consideration both FMN and heme cofactors (Figure 4A–C and Appendix A). The model that features an I-SC score of 35 was also validated using MolProbity (score of 1.6) and fit the higher (score of 3) CAPRI class level for complexes. Interaction energy between domains was estimated in the range of 10 kcal/mol using the Prodigy software. This is very similar to the energies/dissociation constants described previously for CPR with CYP1A1 [51] as well as with CYP2C19, 2D6 and 3A4 [52]. Residues involved in side chain contacts at the interface formed a dense double layer of interacting side chain atoms that could be involved in propagating conformational information from the FMN to the heme cofactor through the interface. The complex model described in Figure 4A,B was used to determine residues at the interface for which significant side chain rearrangement occurred during complex formation. Such residues are particularly prone to inducing conformational information exchanges between the FD and the CYP1A2 domains. Briefly, the best modeled complex was submitted to a series of aggressive relaxations, to generate a collection of conformational variants. Each of these complexes was then split into un-complexed CYP1A2 and FD domains without altering coordinates. Resulting sets of complexed or isolated domains were then submitted to additional and independent relaxation steps. All generated structures were matched to their C-alpha via RMSD minimization and characterized for average side chain geometry differences.

Figure 4D (for the FD) and Figure 4E (for CYP1A2) present the differential RMSD of side chain residues between the complexed and free domains corrected by the intrinsic side chain RMSD of relaxed individual domains or complexes. Such calculation and averaging on a large number of independent relaxations allowed us to remove most of the statistical noise and to evidence positions where a strong side chain-induced fit occurred upon complex formation. A comparison of the positions of contacting residues at the interface and detected induced fit positions clearly illustrated a strong correlation validating the approach. The number of induced fit positions appeared more limited on the CYP side compared to that on the FD. Most interestingly, a large part of the residues involved in a strong induced fit on the FD are identical or in close contact in the structure with several of the described mutations modifying the profiles of caffeine metabolites (Figure 2). This is particularly the case for mutations G175D (three positions), G144C, T142A (two positions), H183Y and P117H (three positions).

### 3.7. Caffeine Docking to CYP1A2

To understand how caffeine binding could affect the CYP1A2 structure, in silico binding studies were performed. First, the α-naphthoflavone molecule was removed from the CYP1A2 crystal structure (PDB code 2HI4) [53]. After relaxing the resulting structure in water using Yasara (see Material and Methods), docking experiments (a total of 300) were performed with caffeine, resulting in 138 distinct clusters. An analysis of these clusters demonstrated that the first 12 corresponded to caffeine bound within the active site, in the distal cavity of the heme. In these clusters, only four caffeine orientations could be found. The three docked structures issued from the structures with the lowest interaction energies (thus probable best binding) and corresponding to the putative orientations of the three major caffeine products with CYP1A2 are presented in Figure 5A. These were designated as the first, second and third position, respectively, and are compatible with the formation of theophylline, paraxanthine and theobromine, respectively, in concordance with what has been previously described [54]. A fourth orientation (Appendix A), compatible with C8 hydroxylation, was not used for further analysis, as this fourth metabolite (1,3,7 trimethyluric acid, a very minor (<1%) metabolite) was not detected under our experimental conditions.

### 3.8. Molecular Dynamics of Caffeine-Free or Caffeine-Bound CYP1A2

MD simulations of the three different caffeine-bound structures as well as of CYP1A2 devoid of any substrate were performed (for 25 ns; four replicas for each simulation). All simulations reached equilibrium and the overall RMSD for each simulation reached an average of 1.5 Å for backbone atoms and 2 Å for all atoms in a relatively short time (approximately 2.5 ns). This demonstrated adequate levels of stabilization (Appendix A), albeit with minor modifications of the overall RMSD in some runs. We then analyzed the 16 different simulations, specifically looking at possible variations between the caffeine-free structure and the caffeine-bound ones for all residues and then specifically concentrating on the residues that were found to be involved in the interface with the FD of CPR (see Figure 4A–C, and Appendix A). RMSF differences between caffeine-bound and caffeine-free simulations were assessed. This analysis was performed for all residues (Appendix A) or restricted to residues at the interface (Figure 5B,C). Interestingly, the plots show differences between each of the three caffeine-bound positions as well as differences when looking at backbone or all atoms’ RMSF values. We hypothesize that the different binding poses of the caffeine molecule induce differential effects on the dynamics of specific residues present at the interface. These differences can affect only the backbone atoms, or all atoms, and can either show higher or lower RMSF values than those of the caffeine-free simulations. While analyzing each residue would have been quite cumbersome, the full RMSF values for some specific residues illustrate specific differences compared to no difference at all (see Appendix A). Residue Q364 showed very minor differences while the three others showed differences between caffeine-free and caffeine-bound simulations in the first and second position (E446), between caffeine-free and caffeine-bound simulations in the second position (F147) and between caffeine-free and caffeine-bound simulations in the third position (P295). Therefore, differences in RMSF values can be seen between caffeine-free and caffeine-bound simulations and these differences seem specific to the binding orientation. Hence, the positioning of caffeine in the active site influences the dynamics of both backbone atoms and all atoms of the residues at the interface. These results indicate that the three binding poses of caffeine induce different subtle but measurable changes in the dynamics of specific residues present at the CPR binding surface of CYP1A2.

## 4. Discussion

CYP1A2 metabolizes 95% of the ingested caffeine and is the unique isoform responsible for paraxanthine formation, in addition to having the capacity to produce theophylline and theobromine. CYP2E1 can catalyze only the formation of theophylline and theobromine, accounting only for 5% of caffeine metabolism [29,48]. This enables the use of caffeine as a probe substrate for CYP1A2 phenotyping [28]. As such, CYP1A2-mediated caffeine metabolism was applied as a model reaction to study the role of the protein dynamics of CPR in the specificity of CYP metabolism, using 14 different CPR variants previously demonstrated to affect the open/close CPR dynamics and/or the CYP binding at the FD interface.

Membrane fractions containing CYP1A2 and CPRwt demonstrated the formation of N1, N3 and N7 demethylation products, with a stoichiometry very similar to that described previously [29,48]. 1,3,7 trimethyluric acid (TMU), a C8 hydroxylation product previously shown as a very minor fourth primary CYP1A2 metabolite [55,56], was not detected by us. However, these studies made use of recombinant human CYP1A2, detecting TMU only when applying either very high caffeine concentrations (25 mM) and/or non-physiological CPR:CYP1A2 ratios (>1). In this current study, a much lower caffeine concentration (600 µM) was used, and additionally, our CYP1A2/CPR-containing membrane fragments demonstrated a physiologically relevant sub-molar co-expression of CPR with CYP. CYP1A2 may act on the primary metabolites paraxanthine or theophylline, generating the secondary caffeine metabolites 1-methylxanthine and 3- methylxanthine, respectively [29]. Both these metabolites were not detected using our experimental approach.

Several of the 14 tested CPR variants demonstrated differences in caffeine consumption velocities, when compared to the CPRwt, indicating altered efficiencies for these variants in caffeine metabolism (Figure 2). These results confirm that substrate binding by CYPs modulates electron transfer properties from CPR, as previously reported by us [19,21,23] and others [57]. Several mutants demonstrated altered patterns in the formation of the metabolite profiles (Figure 2), compared to those of the CPRwt, particularly FD mutants T142A, V164M, H183Y and A229T and the three hinge mutants S243P, I245P and R246A, foremost detected in the alteration of the two minor metabolites theobromine and theophylline. These results indicate that in fact modification of the hinge segment (modifying the conformational equilibrium between closed and open conformers of CPR) as well of the FD (modifying the interaction between CPR and its redox partners) can modulate the specificity of CYP-mediated reactions. Of importance, two CPRvar/1A2 couples containing natural FD variants demonstrated a significant increased production of theophylline (T142A, allele *POR**12 [58]) and theobromine (V164M, *POR**45 [59]). This is indicative that natural variants of CPR may induce differences in caffeine metabolism mediated by CYP1A2, not only modulating reaction efficiency (as we demonstrated before for other substrates [31,60]) but also altering reaction product specificities. The effect of surrogate redox partners on velocity and metabolic outcome has been demonstrated for bacterial and yeast CYPs (reviewed in [61,62]). However, to our knowledge, the effect on metabolite specificity has not been shown for mammalian CYPs and their redox partners. This is of particular relevance given that the genetic polymorphism of these partners (such as human *POR* variation) seems to be part of this phenomenon.

Different experimental approaches were used to clarify the observed differences in CYP1A2-mediated caffeine metabolism and unravel the possible molecular mechanisms of these effects. Uncoupling leading to ROS formation (presented in Appendix A) was generally not increased as compared to the wild-type enzyme by CPR mutations except for T142A and A229T. However, in these cases, ROS formations remained marginal in comparison to caffeine consumption even considering that ROS absolute values could be underestimated due to the documented ROS chemical scavenging mediated by caffeine [63]. The lack of correlations between uncoupling, regiospecificity modulation and caffeine consumption in regard to CPR mutations could appear surprising but might indicate that CPR electron transfer rates to the oxidized and/or the ferrous dioxygen P450 does not constitute the catalytic limiting step in our experimental conditions. Such assumption is also supported by the presented independence of caffeine turnover on the CYP to CPR ratios. It appears in such conditions that conformational coupling of CPR mutations to catalytically determining CYP structural elements more likely constitute a determining factor of our observations.

Even though T142 was not found within the group of residues having the greatest RMSF differences in the CPR–CYP1A2 interface (Figure 4D), this residue has been identified in humans with perturbed steroidogenesis induced by *POR* deficiency [6]. However, we note that T142 is close to residues E145 and D147, which are both part of a negatively charged interaction patch on FD [21]. We suggest that the mutation of a threonine to an alanine at position 142 may induce conformational changes affecting acidic residues E145 and D147 (and potentially even D150 and D154), thereby impacting the electron transfer of FD to CYP.

While it would be tempting to draw conclusions on the conformational role of specific residues affected by caffeine and present at the interface (Figure 5B), it should be noted that these simulations were performed with oxidized CYP1A2, with a 25 ns timescale and in the absence of the membrane-binding region. Although the list of affected residues will certainly show deviations when performing a more exhaustive MD simulation, the binding of caffeine conveys structural changes to the interface between CYP1A2 and FD. Moreover, the three poses of caffeine, compatible with the formation of the three detected metabolites, seem to relay pose-specific structural changes and thus impact differentially the residues in the interaction area between CYP1A2 and FD. As demonstrated, the formation of the complex between the FD and the CYP caused a significant number of side chain geometry changes on both proteins. The interface forms a dense and continuous double layer of both intradomain and interdomain interactions prone to transmitting conformational coupling to surrounding residues. These may include residues lining the active site, as previously reported for the interaction of FD with CYP17A1 [64], and may even reach the heme cysteine ligand, as was recently described for CPR’s interaction with the very closely related CYP1A1 [51]. Our data seem to indicate that differential substrate orientation in the CYP’s active site allosterically modulates its proximal surface in ways that have differential interactions with reductase mutants. That is, proximal CYP1A2 conformations induced by a caffeine pose may preferentially bind a particular CPR variant, with improved ET and catalysis, generating the corresponding product. Conversely, a CPR variant with an altered CYP-binding surface and/or open/closed dynamics may interact less efficiently with CYP1A2, generating smaller amounts of that particular product. Both will lead to altered metabolite profiles. Nevertheless, our modeling data cannot exclude CPR mutations imposing preferences on substrate binding poses, indicating a bidirectional effect of substrate and redox partner binding on CYP catalysis, as suggested very recently by others [65]. At least five of the FD mutations (P117H, T142A, G144C, G175D and H183Y) which demonstrated altered specificity in caffeine metabolism are prone to conformational coupling with residues involved in the induced fit binding to CYP1A2 (Figure 4D). Each of these five FD variants may thus mediate, in a variant specific manner, changes in the interaction domain. Such coupling effects can propagate up to the heme binding region and result in the conformational selection of caffeine binding modes. The three studied hinge mutants also demonstrated deviated metabolite patterns. Although no modeling of these variants was performed due to computational constraints, we can hypothesize that these mutations, by altering the equilibrium between open and closed conformers, may very likely modulate the allosteric effects in the CPR–CYP1A2 interaction.

## 5. Conclusions

In summary, our study demonstrated that both the hinge segment (responsible for the aggregate of open conformers of CPR) as well the FD (the interaction domain of CPR for the redox partner) play a role in the specificity of CYP1A2-mediated reactions, which can be altered when these two structural features are modified. Naturally occurring variations of CPR (implicating with both the hinge segment and FD) have been shown to impact xenobiotic metabolism (decreasing or accelerating it) in an apparent CYP isoform-dependent manner [6]. However, CPR variability in the specificity of metabolite formation by CYP reactions has so far not been shown. CYP1A2 is involved in the bioactivation of several human procarcinogens (e.g., aromatic and heterocyclic amines [1]) but also of therapeutic drugs such as clozapine. This drug, an antipsychotic used in the treatment of refractory schizophrenia with frequent severe side effects, is mainly metabolized by CYP1A2, resulting in two metabolites with different toxicological and pharmacological profiles [66]. CPR variability seems likely to be an additional factor in the observed variability in therapeutic outcomes and adverse effects of this drug. Based on the common molecular mechanism of ET by CPR to all microsomal CYPs, it is very plausible that CPR variability may impact other CYPs in their metabolic specificity, in addition to CYP polymorphism.

## Figures and Tables

**Figure 1 biomolecules-13-01083-f001:**
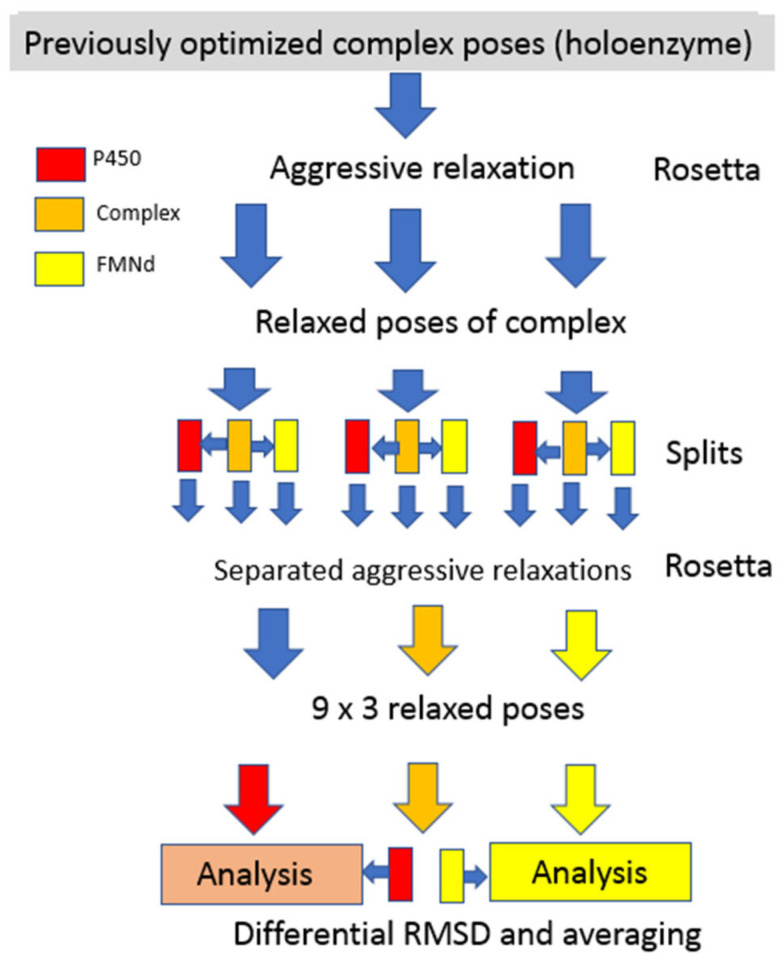
Flow chart for the determination of induced fit RMSD.

**Figure 2 biomolecules-13-01083-f002:**
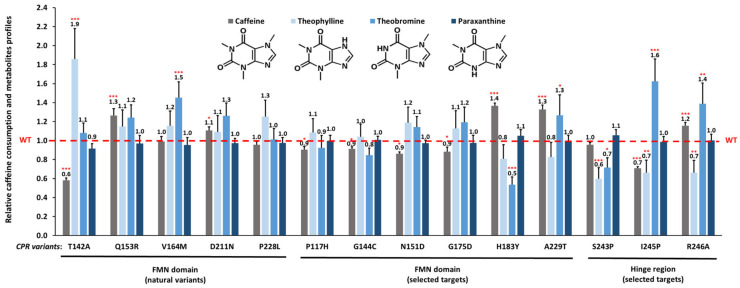
Caffeine consumption activity and metabolite profiles produced by CYP1A2 in combination with the 14 CPR variants. Relative caffeine consumption rates (*k*_obs_) of the CPRvar/1A2 couples, normalized by the caffeine consumption rate (*k*_obs_) demonstrated by CPRwt/1A2 (grey bars). Metabolite (theophylline, theobromine and paraxanthine) profiles (blue bars) of the CPRvar/1A2 couples were determined relative to the metabolites formed by the CPRwt/1A2 (i.e., metabolite frequency of CPRvar-1A2 couples divided by metabolite frequency of CPRwt/1A2 couples. (WT: CPRwt/CYP1A2, dashed red line (technical replicates N = 3; *** *p* < 0.001; ** *p* < 0.01; * *p* < 0.05)).

**Figure 3 biomolecules-13-01083-f003:**
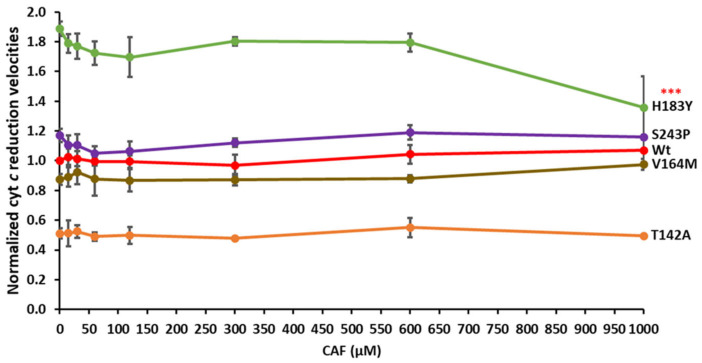
Cytochrome *c* reduction rates of four selected CPR variants, plotted in function of caffeine concentration. Observed rate constants of cyt *c* reduction (*k*_obs_) (x fold) of the CPRvar/1A2 couples, normalized according to CPRwt/1A2 activity without caffeine (*k*_obs_ (0 nM CAF) (CAF: caffeine) (technical replicates N = 3; *** *p* < 0.001)).

**Figure 4 biomolecules-13-01083-f004:**
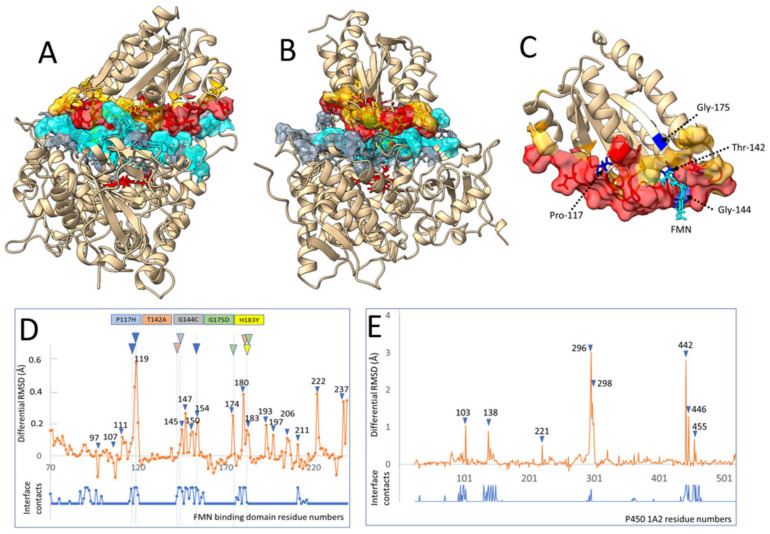
Interaction between CYP1A2 and FD. (**A**,**B**) Views of the interface between CYP1A2 and the FD of the human reductase in two perpendicular orientations. Contacting residues (less than 3 Å) in more than 5 poses over 15 of alternatively modeled relaxed complexes and their surfaces are colored in blue and red for the CYP and the FMN domains, respectively. Less frequent contacts (1 to 4 over 15) in complexes are colored in grey and yellow, respectively. (**C**) Enlarged structure of the FD using the same color code. Five of the considered mutated residues interacting with interface are colored in dark blue and labeled as well as was the FMN cofactor. (**D**,**E**) Induced fit differential RMSD of FD side chains during binding to CYP1A2′s proximal side calculated (orange plot) as described in the experimental section. On top, a color code is given for five of the considered FD mutants examined. This code is reported in colored triangles at the corresponding amino acid positions. When multiple triangles are shown for a given mutant, alternate positions point to other residues that could be interacting with the mutated residue in the 3D structure. The number of interdomain contact hits among the 15 examined complex poses is overlaid (blue trace, see also Appendix A).

**Figure 5 biomolecules-13-01083-f005:**
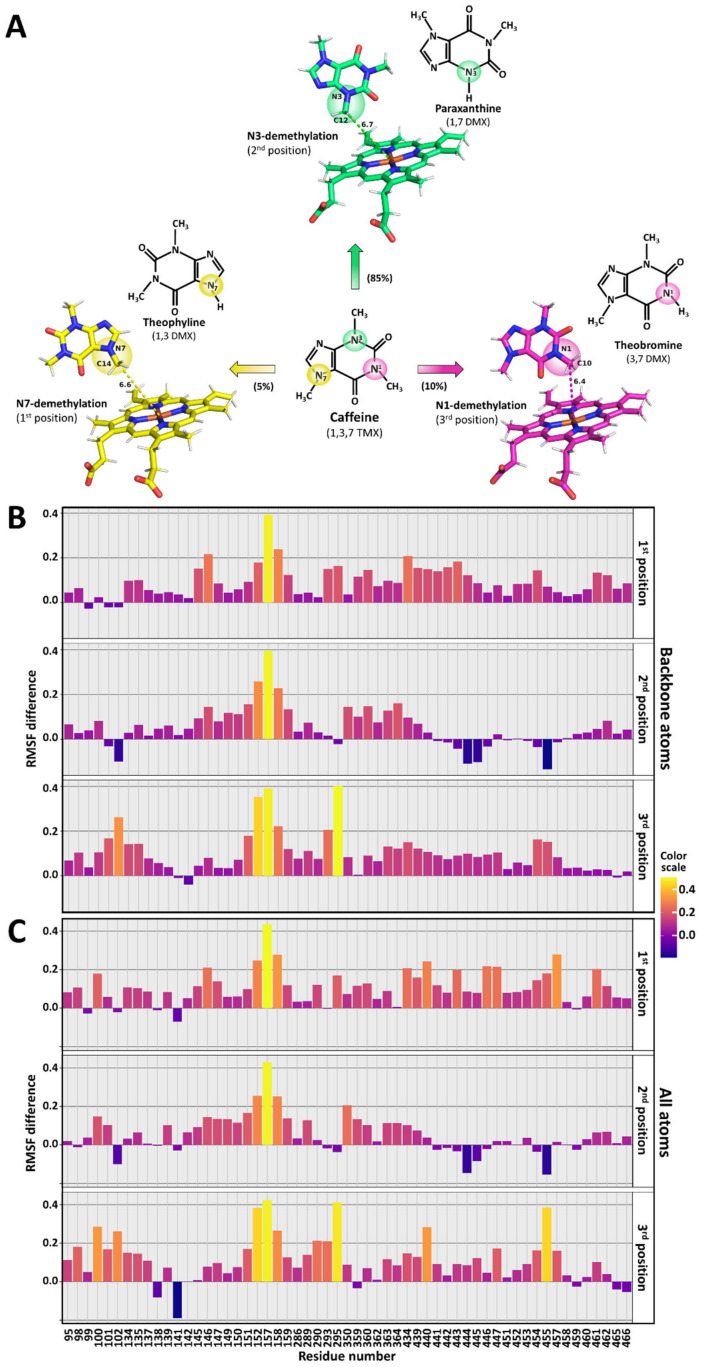
Caffeine binding by CYP1A2. (**A**) Caffeine orientations obtained from different docking clusters. These orientations are compatible with the major N1, N3 and N7 demethylation products: theophylline (yellow structure), paraxanthine (green structure) and theobromine (magenta structure). The heme and caffeine structures are depicted as sticks, along with the distance from the heme iron to the methyl group that was abstracted during catalysis. (**B**,**C**) Difference in RMSF values between caffeine-bound simulations (1st, 2nd or 3rd position) and caffeine-free simulation for the residues at the interface between CYP1A2 and FD, (**B**) backbone atoms, and (**C**) all atoms. The color scale is depicted on the right and corresponds to the difference in RMSF values.

**Table 1 biomolecules-13-01083-t001:** Characterization of BTC1A2 membrane fractions (CYP and CPR contents, protein stability, caffeine consumption and amount of metabolites produced).

Membrane Fractions		Protein Contents			Protein Stability	Caffeine		Metabolites	
CPR Region	CPR Form	CYP 1A2	CPR	CPR/CYP	Relative Cyt *c* Reduction Rates	Consumption	Theophylline	Theobromine	Paraxanthine
		(pmol/mg Protein)		Ratios	(6 h/0 h)	(µM)			
FMN domain (natural variants)	T142A	145 ± 2	38.5 ± 0.9	1:4	1.01 ± 0.13	5.82 ± 0.18	0.85 ± 0.10	0.39 ± 0.01	4.58 ± 0.15
	Q153R	267 ± 3	26.0 ± 0.4	1:10	0.85 ± 0.12	12.96 ± 0.66	1.16 ± 0.08	1.00 ± 0.04	10.80 ± 0.66
	V164M	259 ± 2	35.2 ± 0.8	1:7	0.94 ± 0.16	10.15 ± 0.48	0.92 ± 0.02	0.91 ± 0.05	8.32 ± 0.48
	D211N	211 ± 4	24.9 ± 0.7	1:8	1.03 ± 0.16	11.34 ± 0.31	0.97 ± 0.09	0.89 ± 0.05	9.49 ± 0.30
	P228L	451 ± 4	41.9 ± 1.1	1:11	0.95 ± 0.14	9.57 ± 0.30	0.94 ± 0.05	0.60 ± 0.04	8.03 ± 0.28
FMN domain (selected targets)	P117H ^a^	148 ± 1	12.2 ± 0.5	1:12	0.93 ± 0.14	9.29 ± 0.30	0.79 ± 0.03	0.53 ± 0.06	7.97 ± 0.30
	G144C ^a^	124 ± 2	14.1 ± 0.2	1:12	0.93 ± 0.17	9.39 ± 0.11	0.76 ± 0.03	0.49 ± 0.01	8.14 ± 0.10
	N151D ^a^	446 ± 4	27.7 ± 0.5	1:12	1.02 ± 0.13	8.85 ± 0.07	0.82 ± 0.50	0.63 ± 0.02	7.40 ± 0.05
	G175D ^a^	379 ± 2	26.5 ± 0.4	1:14	1.01 ± 0.05	9.09 ± 0.45	0.80 ± 0.08	0.67 ± 0.05	7.62 ± 0.45
	H183Y	179 ± 2	14,9 ± 0.3	1:12	0.89 ± 0.06	14.05 ± 0.08	0.89 ± 0.12	0.47 ± 0.06	12.70 ± 0.64
	A229T ^a^	71 ± 4	17.3 ± 0.2	1:4	0.99 ± 0.11	13.30 ± 0.28	0.86 ± 0.12	1.04 ± 0.15	11.40 ± 0.52
Hinge region (selected targets)	S243P ^a^	73 ± 4	7.7 ± 0.2	1:9	0.60 ± 0.06 **	9.81 ± 0.06	0.46 ± 0.07	0.44 ± 0.05	8.92 ± 0.40
	I245P ^a^	102 ± 1	6.1 ± 0.5	1:17	0.98 ± 0.12	7.28 ± 0.09	0.38 ± 0.06	0.73 ± 0.08	6.17 ± 0.28
	R246A ^a^	91 ± 2	5.4 ± 0.2	1:17	0.84 ± 0.06	11.89 ± 0.23	0.62 ± 0.09	1.02 ± 0.13	10.25 ± 0.54
-	Wt (A)	56 ± 3	23.3 ± 1.6	1:2	0.93 ± 0.05	10.02 ± 0.28	0.77 ± 0.09	0.63 ± 0.05	8.61 ± 0.08
-	Wt (B)	116 ± 3	12.4 ± 0.8	1:9	0.88 ± 0.07	10.25 ± 0.23	0.80 ± 0.10	0.63 ± 0.05	8.81 ± 0.20
-	Wt (C) ^a^	54 ± 1	4.1 ± 1.5	1:13	0.84 ± 0.11	10.28 ± 0.21	0.82 ± 0.10	0.63 ± 0.06	8.83 ± 0.20
-	CPRnull ^a^	85 ± 3	ND	-	ND	ND	ND	ND	ND

^a^ Membrane fractions from batches used in our former studies [19,21,23]. Wt (A) was used as reference for T142A and A229T, Wt (B) was used as a reference for Q153R, V164M, D211N, P228L and S243P, and Wt (C) was used as a reference for P117H, G144C, N151D, G175D, H183Y, I245P and R246A, in determining the relative activities, metabolite profiles, and relative cyt *c* reduction rates. Primary caffeine CYP1A2-metabolite 1,3,7 trimethyluric acid and secondary metabolites 1-methylxanthine and 3-methylxanthine were not detected. CPR protein stability was assessed by determining the ratios of cyt *c* reduction (Abs/min) measured after a 6 h incubation period relative to the ones obtained at the initial time point (0 h). (ND) not detected. Values are mean ± SD (technical replicates N = 3) (** *p* < 0.01).

## Data Availability

All data are included in this manuscript or the Appendix A. All modeling data are available upon request.

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
