# Peer review of "Single Mutations in Cytochrome P450 Oxidoreductase Can Alter the Specificity of Human Cytochrome P450 1A2-Mediated Caffeine Metabolism"

_biomolecules, 2023, doi:10.3390/biom13071083_

Round 1

Reviewer 1 Report

This intriguing report of mutations in the NADPH-cytochrome P450 reductase hinge and FMN domain affecting the regiospecificity of human cytochrome P450 1A2 is highly intriguing.  The solid enzymatic results shifting metabolite production is then followed by a more speculative computational attempt to understand the protein-protein interaction that is thought provoking.  

The authors show data effectively correlating reductase or FMN domain mutations with the formation of different products from the CYP1A2 active site.  They suggest that differential reductase/P450 interactions allosterically cause different orientations of the active site substrate, which seems a potentially valid hypothesis.  Is there evidence that it could work the other way around?  That differential substrate orientation in the P450  active site allosterically modulates the P450 proximal surface in ways that have differential interactions with reductase mutants; i.e. some proximal CYP1A2 conformations bind a particular reductase mutant better (and so e- transfer and catalysis occurs more readily, generating the corresponding product) or worse (so less of that product is generated)?

I think the computational approach would be more clearly communicated if Figure S1 was moved to the main text.

In Figure S2 it is not easy to see where the mutations are located.  Perhaps add an arrow or circle to help readers identify them?

Finally, aren't there only 48 human P450 enzymes employing reductase?  I agree that 50 are microsomal but CYP5A1 and CYP8A1 don't use reductase, at least not for their major activities.

• Misspellings: substrate caffeine (several times, check throughout); methoxyresorufine->methoxyresorufin

• A couple of word choice concerns:  "enzyme superfamily" and "donation capacitates microsomal"->"donation supports microsomal"; "of de FAD"; "seem to characterize"->"seem to support"; "CYP's substrate binding site is highly malleable" probably mostly applies to drug-metabolizing CYPs and less so to ones in endogenous pathways, so suggest "Many CYPs"?; 807:  "tinkered"->"altered"

* Methods, line 2067:  "conditions were"; 211: "with the selected temperatures"' 300: "orientations yielding the three major"

• Results:  376:  G144C; a line or two seems to be missing under figure 3

• RMSF doesn't appear to ever be defined; neither are the compounds in Table S2 or PhAC in supplemental

Author Response

  1. the issue raised if the “differential reductase/P450 interactions allosterically cause different orientations of the active site substrate” may work the other way around we have incorporated this suggestion; see lines 799-809
  2. the computational approach, and Supplemental figure we have now placed Figure S1 in the main text body (now Figure 1) of the manuscript, see page 7
  3. the issue of CYP5A1 and 8A1 not being exclusively dependent in their activity on CPR we have adapted the text: see lines 15 and 38.
  4. All misspellings and suggestions were addressed.

Reviewer 2 Report

The work by Esteves et al. presents results of investigation of caffeine demethylation reaction by cytochrome P450 1A2 in the presence of cytochrome P450 reductase containing various mutations. The study indicates that certain mutations affect not only rate of caffeine demethylation, but also the composition of the products. In general, this is the thoroughly done study with noteworthy findings, and it is suitable to publication in Biomolecules. However, I recommend to pay attention to following points:

- Fig. 2: data for H183Y exhibits no descending profile up to 0.6 mM caffeine concentration, whereas slight inhibition is observed only at 1 mM. It is very risky to make conclusions only using the results obtained for the single concentration. Additional data for adjacent caffeine concentrations (e.g., 0.8 and 1.2 mM) can significantly enhance the conclusions.

- legend for Fig. 3 is damaged.

- indicate ROS generated upon caffeine processing by CPR/CYP. Apparently, the process involves formation just some of them, i.e., of hydrogen peroxide, superoxide (?), etc.  

- Schemes illustrating mechanisms of the processes are very welcome. In particular, for an inexperienced reader, it may not be clear how theophylline and other species are formed from caffeine upon processing by CYP.

Author Response

  1. Re the fact that variant H183Y no showing any descending profile up to 0.6 mM, we have altered our conclusion and adapted the text. See lines 452-454, and 779-783.
  2. Fig 3 (now Figure 4, page 15) has been reallocated, in such way that the legend is not covered.
  3. The raised issue re ROS species has now been clarified, by the describing text, re the 2′,7′-dichlorofluorescein assay, see lines 184 and 429-430.
  4. The issue raised re. the necessity of a scheme explaining the formation of the three metabolites seems a little odd to the authors. The formation of the three metabolites via demethylation at the N1, N3 and N7 position of caffeine leading to the metabolites, is described in detail, including chemical structures and N-numbering, in Figure 4A (now Figure 5A). Adding a reaction scheme of caffeine leading to these three metabolites seem to us overlapping with what is presented in Figure 4A (now 5A).